# R-loop resolution promotes co-transcriptional chromatin silencing

Congyao Xu [1,5], Zhe Wu[1,2,5], Hong-Chao Duan [3,5], Xiaofeng Fang[1,4], Guifang Jia [3✉] & Caroline Dean [1✉]

RNA-mediated chromatin silencing is central to genome regulation in many organisms. However, how nascent non-coding transcripts regulate chromatin is poorly understood. Here, through analysis of Arabidopsis *FLC*, we show that resolution of a nascent-transcript-induced R-loop promotes chromatin silencing. Stabilization of an antisense-induced R-loop at the 3′ end of *FLC* enables an RNA binding protein FCA, with its direct partner FY/WDR33 and other 3′-end processing factors, to polyadenylate the nascent antisense transcript. This clears the R-loop and recruits the chromatin modifiers demethylating H3K4me1. FCA immuno-precipitates with components of the $m^6A$ writer complex, and $m^6A$ modification affects dynamics of FCA nuclear condensates, and promotes *FLC* chromatin silencing. This mechanism also targets other loci in the Arabidopsis genome, and consistent with this *fca* and *fy* are hypersensitive to a DNA damage-inducing drug. These results show how modulation of R-loop stability by co-transcriptional RNA processing can trigger chromatin silencing.

[1] Cell and Developmental Biology, John Innes Centre, Norwich, UK. [2] Key Laboratory of Molecular Design for Plant Cell Factory of Guangdong Higher Education Institutes, Institute of Plant and Food Science, School of Life Sciences, Southern University of Science and Technology, Shenzhen, China. [3] Synthetic and Functional Biomolecules Center, Beijing National Laboratory for Molecular Sciences, Key Laboratory of Bioorganic Chemistry and Molecular Engineering of Ministry of Education, College of Chemistry and Molecular Engineering, Peking University, Beijing, China. [4] Center for Plant Biology, School of Life Sciences, Tsinghua University, Beijing, China. [5]These authors contributed equally: Congyao Xu, Zhe Wu, Hong-Chao Duan. ✉email: guifangjia@pku.edu.cn; caroline.dean@jic.ac.uk

Antisense transcription is pervasive in many genomes, although the transcripts are rarely detected due to their rapid turnover[1]. This has made the analysis of antisense transcript function difficult to elaborate. A classic example of antisense-mediated chromatin silencing is at the floral repressor locus, Arabidopsis *FLC*. The antisense transcripts, named *COOLAIR* through their induction by cold temperatures[2], are also functionally important at warm temperatures as part of the autonomous floral pathway[3]. The autonomous pathway transcriptionally silences *FLC*, and thus accelerates flowering. Genetic screens have identified RNA-binding proteins (FCA, FPA, and FLK), conserved 3′-end processing factors (FY/WDR33, CstF64, CstF77), the PRP8 splicing factor, and FLD a histone demethylase homolog, as components necessary for this transcriptional silencing[3–6]. These co-transcriptional regulators promote proximal polyadenylation of *COOLAIR*[7–9], which leads to the recruitment of a set of physically interacting chromatin modifiers FLD/LD/SDG26[10]. These inhibit H3K4me1 and H3K36me3 accumulation and by antagonizing transcription, promote H3K27me3 accumulation, reducing *FLC* transcriptional initiation and elongation rates. Previously, we have reported an R-loop (three-stranded nucleic acid structure, RNA hybridized to DNA with a displaced single-stranded DNA) is generated by the *COOLAIR* transcript at the 3′-end of *FLC*[11]. Genomic analysis indicates a link between R-loop accumulation and certain histone modification[12,13], with links to chromatin silencing[14–17], however, how R-loops influence chromatin silencing is still unclear. Here, we describe a mechanism in which modulation of R-loop stability by co-transcriptional RNA processing involving m[6]A modification can trigger chromatin silencing.

## Results

### Stabilization of the *COOLAIR* R-loop promotes *FLC* chromatin silencing

The *COOLAIR* R-loop at the 3′-end of *FLC* is stabilized by the homeodomain protein AtNDX, which inhibits further antisense transcription[11]. The R-loop corresponds in length to the nascent short proximal *COOLAIR* (~626 nucleotides) (Fig. 1a), whose formation is promoted by the RNA-binding proteins FCA and FPA and the canonical 3′-end processing factors FY, CstF64, and CstF77[7,18]. To address how NDX and FCA function together mechanistically at *FLC*, we analyzed the genetic interaction between *fca* and *ndx*. *FLC* levels are slightly de-repressed in *ndx1-4*, significantly de-repressed in *fca-9*, and similar to *fca-9* in the double mutant (Fig. 1b and Supplementary Fig. 1), consistent with NDX facilitating *FLC* epigenetic silencing via FCA. FCA associates along the length of the nascent *COOLAIR* transcript (Fig. 1c) and promotes proximal polyadenylation of *COOLAIR*[19]. However, this association was found to be reduced in *ndx1-4* (Fig. 1c), which together with reduced levels of the R-loop[11], suggests that R-loop stability influences FCA–*COOLAIR* association. We, therefore, asked whether the reduced FCA–*COOLAIR* association would affect *FLC* chromatin silencing and found that H3K4me1, a modification we have recently shown needs to be removed to establish a H3K27me3 domain at *FLC*[19], accumulates over *FLC* in *ndx1-4*, although to a lesser extent than *fca-9*

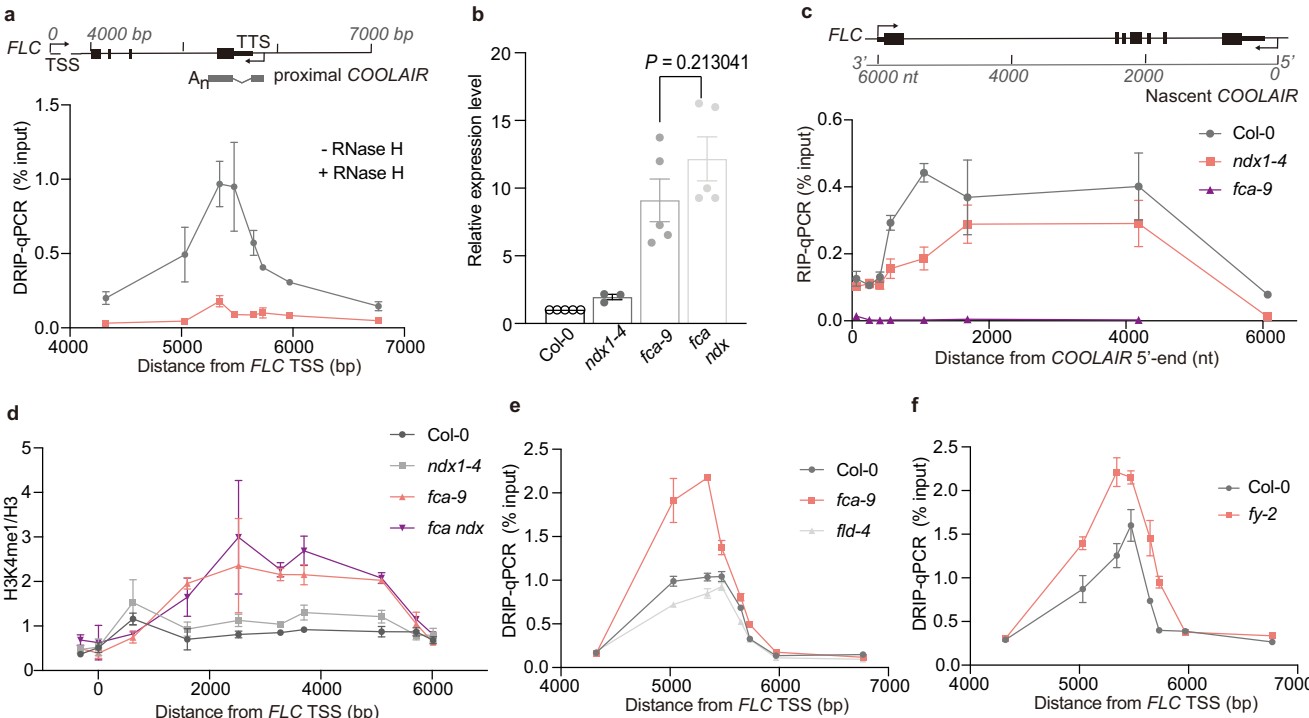

**Fig. 1 Co-transcriptionally formed R-loop promotes *FLC* chromatin silencing. a** S9.6-DNA/RNA immunoprecipitation (DRIP)-qPCR analyzing the R-loop over 3′-end of *FLC* in wild-type Col-0, with and without RNase H treatment. The number on x axis is the distance to *FLC* transcription start site (TSS = 0), and x axis is corresponding to the schematic on the top. TTS transcription termination site. Data are mean ± s.e.m. from three independent experiments. **b** Expression of spliced *FLC* relative to *UBC* in various genetic backgrounds. Data are normalized to wild-type Col-0. Data are mean ± s.e.m. from three to five biological replicates. Two-tailed P value from multiple t test corrected by Holm–Sidak method. **c** FCA-RIP-qPCR analyzing FCA enrichment on *COOLAIR* transcript in Col-0, *ndx1-4* and *fca-9* (negative control). The number on x axis is the distance to *COOLAIR* 5′-end. Data are mean ± s.e.m. from three biological replicates. **d** ChIP analysis of H3K4me1 level at *FLC* in various genetic backgrounds. The number on x axis is the distance to *FLC* TSS. Data are mean ± s.d. from three biological replicates. **e** DRIP-qPCR analyzing the R-loop in Fig. 1a (*COOLAIR* R-loop) in Col-0 and mutants *fca-9* and *fld-4*. Data are mean ± s.e.m. from three biological replicates. **f** DRIP-qPCR analyzing the *COOLAIR* R-loop in Col-0 and *fy-2*. Data are mean ± s.e.m. from three biological replicates. Source data are provided as a Source Data file.

(Fig. 1d). Consistent with *FLC* expression level, the double mutant did not show an additive effect (Fig. 1d). These data imply that the NDX stabilized R-loop structure enhances FCA–COOLAIR association to trigger *FLC* chromatin silencing. R-loop stabilization, potentially with RNA Pol II stalling, may provide an extended time window for FCA to function.

**FCA and FY promote R-loop resolution via efficient 3′-end processing.** Emerging evidence supports the idea that R-loops play regulatory roles in many processes[16,20,21], but accumulated R-loops are harmful to genome stability[22–24]. Therefore, what determines R-loop homeostasis at *FLC* and how FCA might participate in this process were important questions to answer. We found the R-loop level over the *COOLAIR* proximal poly-adenylation site was increased significantly in *fca-9* (Fig. 1e). An unrelated high GC region on a *COPIA* transposon was used as a positive control for R-loop formation and showed no R-loop difference (Supplementary Fig. 2). The increased R-loop abundance was not just a feature of enhanced transcription, as no increase was found in *fld-4* (Fig. 1e), a mutant of the histone demethylase FLD shown to be required for removing H3K4me1 at *FLC*[19], with a similar *FLC* expression level as in *fca-9*. These data also suggest R-loop regulation acts upstream of chromatin modification at *FLC*. Factors in RNA metabolism have previously been shown to prevent R-loop formation[25,26], but our genetic analysis suggests NDX stabilization of R-loop is necessary for FCA action, arguing against FCA preventing R-loop formation. Instead, we considered that FCA, which physically interacts with RNA 3′-end processing factors, might promote efficient 3′-end processing to facilitate R-loop resolution. To test this possibility, we tested the role of FY/WDR33, a canonical 3′-end RNA-processing factor that is a direct interactor of FCA, and is required for FCA function and *COOLAIR* proximal polyadenylation[18]. R-loop levels increased significantly in *fy-2* (Fig. 1f), suggesting that FCA and 3′-end processing factors, which localize into dynamic liquid-like nuclear condensates[19], mediate efficient RNA 3′-end processing to facilitate R-loop resolution.

**m6A modification on *COOLAIR* promotes FCA-mediated *FLC* repression.** Factors co-immunoprecipitated with FCA after formaldehyde cross-linking[19] may function in this FCA/FY-mediated R-loop resolution mechanism. DNA/RNA helicases were identified (Supplementary Table 1), and these might facilitate the unwinding of the RNA/DNA hybrid. Several subunits of the m6A methyltransferase writer complex MTA (homolog of human METTL3), MTB (METLL14), and FIP37(WTAP) were also identified (Supplementary Table 1). The m6A methyltransferase writer complex delivers reversible and dynamic m6A modification to eukaryotic mRNA and lncRNA and is thought to have multiple roles in RNA metabolism and processing[27–31]. To test the functional relevance of m6A methylation in this FCA-mediated mechanism, we carried out genetic analysis by introducing m6A writer mutations into a transgenic line (C2) carrying *35 S::FCAγ* (*FCA* overexpressor) and *FRIGIDA* (encodes an activator of *FLC*) transgenes, a line we had used for suppressor screens to identify components required for FCA-mediated *FLC* repression[7–9,32]. *FLC* is strongly repressed in this sensitized background, which enables us to identify relatively weak-effect mutations that suppress FCA-mediated *FLC* repression. Null mutations of m6A methyltransferases are embryonic lethal, so we used transgenic lines containing embryo-rescued genotypes (*mta* carrying the *pABI3::MTA* transgene[33] and *fip37* carrying the *pLEC1::FIP37* transgene[34]), where m6A modification levels in the seedlings are largely diminished compared to wild-type. Lack of this m6A

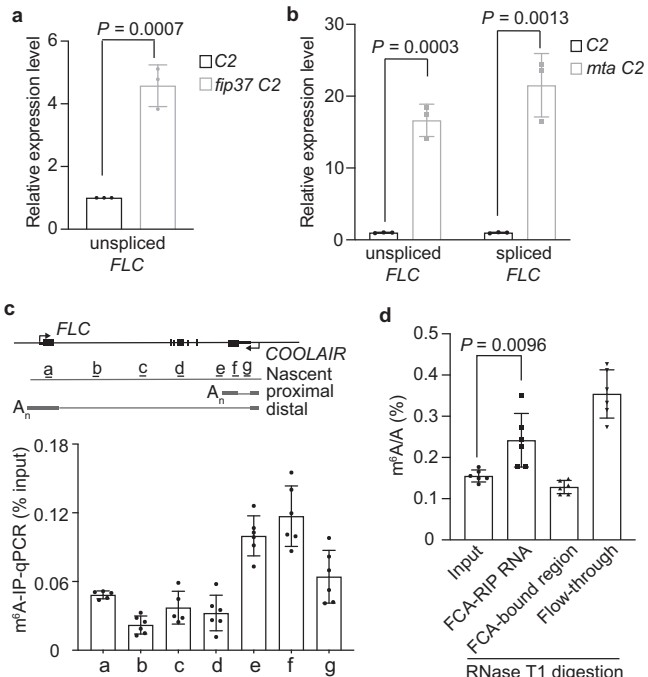

**Fig. 2 m6A methylation on *COOLAIR* is required for FCA-mediated *FLC* silencing. a** Expression of unspliced *FLC* relative to *UBC* in C2 and *fip37* C2 2-week-old seedlings, both carrying transgene *pLEC1::FIP37*. Data are normalized to C2. Data are mean ± s.e.m. from three biological replicates. *P* value from two-tailed *t* test. **b** Expression of unspliced and spliced *FLC* relative to *UBC* in C2 and *mta* C2 (carrying transgene *pABI3::MTA*) plants. Data are normalized to C2. Data are mean ± s.e.m. from three biological replicates. *P* value from two-tailed *t* test. **c** m6A-IP-qPCR analyzing m6A enrichment on chromatin-associated *COOLAIR* transcript after immunoprecipitated by m6A antibody. **a–g** represent amplicons over *COOLAIR* in qPCR analysis, with positions indicated on the top schematic. Data are mean ± s.d. from five to six biological replicates. **d** LC-MS/MS determined m6A-to-A ratio on nuclear RNA immunoprecipitated by FCA, with and without RNase T1 digestion. Data are mean ± s.d. from six biological replicates. Two-tailed *P* value from multiple *t* test corrected by Holm–Sidak method. Source data are provided as a Source Data file.

methyltransferase activity strongly suppressed FCA function resulting in enhanced *FLC* expression (Fig. 2a, b). Consistent with this, *mta* in the C2 background showed a strong late-flowering phenotype (Supplementary Fig. 3a). Importantly, neither FCAγ or endogenous FCA protein levels were affected by the *mta* or *fip37* mutation, respectively (Supplementary Fig. 3b, c). These results were reminiscent of the studies in mammals elucidating the roles of RNA-binding protein and m6A writer complex in m6A-modified *XIST* RNA-mediated gene silencing[35,36]. m6A has been reported to be deposited co-transcriptionally[37], we, therefore, asked whether nascent *COOLAIR* is coated with m6A. m6A was enriched over the region of *COOLAIR* that coincides with the proximal form (Fig. 2c). This suggests that FCA does not directly bind to m6A because FCA associates with most segments of *COOLAIR* nascent transcript (Fig. 1c). To confirm this, we undertook in vivo RNA immunoprecipitation (RIP) to analyze m6A enrichment in RNA fractions bound to FCA. The isolated RNAs were digested into nucleosides and subsequently injected into liquid chromatography–tandem mass spectrometry (LC-MS/MS) for m6A detection (a cartoon description in Supplementary Fig. 4). Surprisingly, we found m6A-modified RNAs were significantly enriched in the FCA–RIP fraction (Fig. 2d), indicating FCA binds to RNAs enriched for m6A. However, after RNase T1

digestion, which leaves only FCA-directly bound regions, the m⁶A enrichment was lost (Fig. 2d), indicating FCA indeed does not bind m⁶A directly. Interestingly, we have found the m⁶A to A ratio of total mRNA is reduced in *fca* mutants (Supplementary Fig. 5a) and the m⁶A level on *COOLAIR* is also reduced in *fca-9* (Supplementary Fig. 5b), suggesting FCA facilitates the deposition of m⁶A on some transcripts, including *COOLAIR*. We also found m⁶A is enriched over the 3′-end of the *FLC* sense transcript, but the level did not change in *fca-9* (Supplementary Fig. 5b), suggesting a parallel FCA-independent pathway. Our data collectively show the interplay between m⁶A and FCA is not linear; FCA facilitates the addition of m⁶A on *COOLAIR* but m⁶A modification is required for FCA-mediated *FLC* silencing.

We next asked whether m⁶A affects FCA association with *COOLAIR* using in vitro and in vivo FCA–RIP assays. The 5′-end of FCA containing the RRM domains was incubated with total nuclear RNA, which had been treated with the m⁶A demethylase FTO[38], removing about 70% of the m⁶A modification (Supplementary Fig. 6a). Loss of m⁶A reduced the ability of FCA (RRM) to immunoprecipitate *COOLAIR* (Supplementary Fig. 6b). Full-length FCA showed the same binding to *COOLAIR* as an FCA-RRM domain, and again this was dependent on m⁶A (Supplementary Fig. 6c). In vivo FCA–RIP analysis in *mta* consistently showed less occupancy of FCA on *COOLAIR* at the regions deposited with m⁶A (Fig. 3a). Nevertheless, we noted

FCA largely remained associated with *COOLAIR* in the *mta* mutant (Fig. 3a). This again argues against m⁶A being part of the direct linear FCA recruitment mechanism, as then we would have expected a reduction in the overall FCA occupancy on *COOLAIR*. Rather it suggests the interaction of FCA with *COOLAIR* is enhanced by the RNA m⁶A modification, possibly through an m⁶A influenced *COOLAIR* structure change, as *COOLAIR* has been shown to be strongly structured[39].

**m⁶A influences the dynamics of FCA nuclear condensates in vivo.** The secondary structure of RNA can influence the properties of liquid compartments[40], so it is pertinent that FCA compartmentalizes with 3′-end processing factors in dynamic liquid-like nuclear condensates both in vitro and in vivo[19]. The phase separation processes involved in the formation of stress granules are enhanced by RNAs containing multiple m⁶A modifications[41]. Both MTA and FIP37 are targeted to nuclear speckles in a transient assay[42]. *Arabidopsis* MTA protein and its orthologs in other plant species (i.e., *G. max, S. tuberosum*) are predicted to be highly disordered and contain a prion-like domain (PrLD) (Supplementary Fig. 7a, b). Interestingly, MTA orthologs in other organisms (i.e., *H. sapiens, D. melanogaster, D. reiro*) are also predicted to be highly disordered but do not contain any PrLDs (Supplementary Fig. 7b). FCA contains two

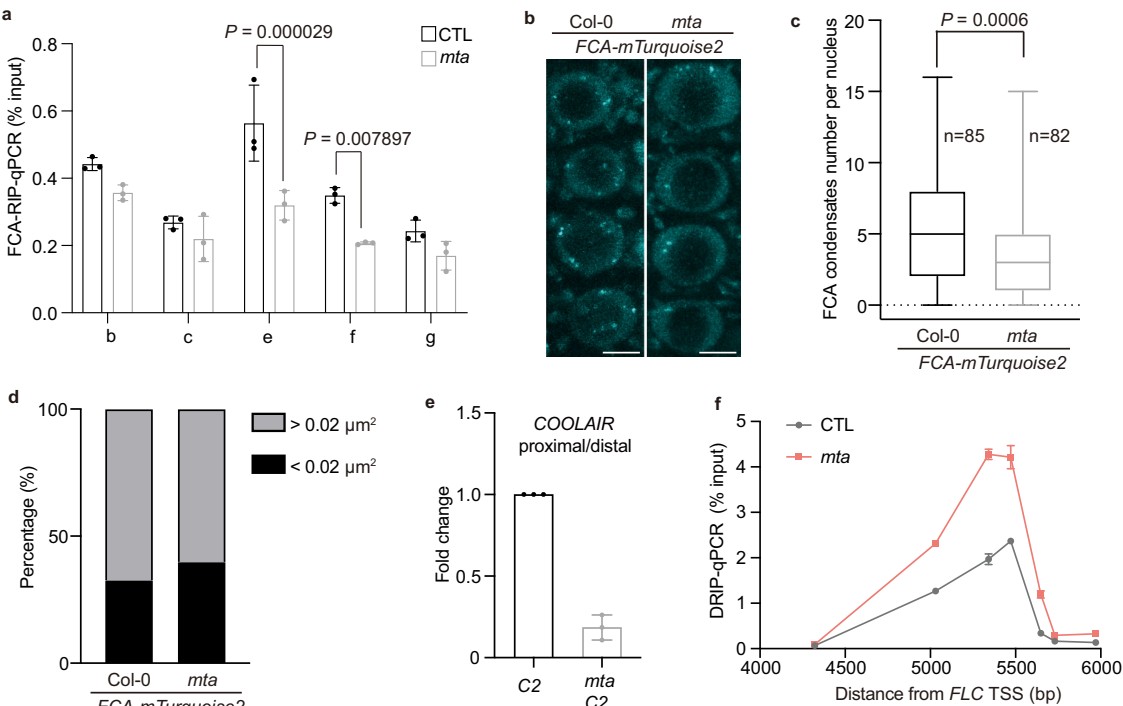

**Fig. 3 m⁶A enhances FCA binding to *COOLAIR* and FCA condensates formation. a** FCA–RIP-qPCR analyzing FCA binding to *COOLAIR* transcript in *mta* mutant and the corresponding wild-type control (CTL). x axis represents the amplicons in qPCR (positions refer to the schematic in Fig. 2d). Data are mean ± s.d. from three biological replicates. Two-tailed *P* value from multiple *t* test corrected by Holm–Sidak method. **b** Representative images of root tip nuclei expressing p*FCA::FCA-mTurquoise2* in plants with and without *mta* mutation. Maximum intensity projections of Z-stack spanning the entire width of a nucleus were applied. Scale bars, 5 μm. **c** Quantification of FCA-mTurquoise2 condensates number in root cells in plants with and without *mta* mutation. Data were plotted from minima to maxima. The box extends from the 25th to 75th percentiles. The line inside the box marks the median. The whiskers go down to the minima and up to the maxima. *P* value from two-tailed *t* test. **d**, The distribution of FCA-mTurquoise2 condensates of different sizes (in two groups, size between 0.01 and 0.02 μm² and bigger than 0.02 μm²) in plants with (*n* = 59 nuclei) and without (*n* = 60 nuclei) *mta* mutation. *P* = 0.0237, two-sided Fisher's exact test. **e** The ratio of proximal-to-distal isoforms of *COOLAIR* transcripts (refer to the schematic in Fig. 2d) in *mta* C2 relative to corresponding wild-type C2. Data are mean ± s.d. from three biological replicates. **f** DRIP-qPCR analyzing *COOLAIR* R-loop in *mta* mutant and corresponding wild-type control (CTL). Data are mean ± s.d. from three biological replicates. Source data are provided as a Source Data file. Raw and processed images for Fig. 3c, d are available from Figshare (https://doi.org/10.6084/m9.figshare.13645730.v1).

predicted PrLDs[19], so we hypothesized MTA and m[6]A deposition might affect the dynamics of FCA nuclear condensates, enhancing the association of FCA and *COOLAIR*. Functionally this would prolong occupancy time of FCA and 3′-end processing factors at sites of nascent *COOLAIR* production and promote R-loop resolution. To test this hypothesis, we undertook an in vivo analysis of the nuclear condensates by imaging root nuclei in transgenic plants. We generated plants carrying an *pFCA::FCA-mTurquoise2* fusion (in a Col genotype), in which the transgene fully complemented the *fca-9* mutation, and crossed to the embryo-rescued genotype *pABI3::MTA* in *mta*, reduced in m[6]A levels in seedlings. In wild-type plant roots, the FCA-mTurquoise2 was localized to nuclear condensates of various sizes (Fig. 3b), similar to the FCA-eGFP fusion which we had reported previously (although this had been in a Ler genotype)[19]. After the introduction of the *mta* mutation, the number of FCA-mTurquoise2 foci reduced from a mean of 5.3 per nucleus to 3.4, and the number of larger condensates reduced from ~67% of the total to ~60% of the total (compared to segregants from the same cross, guaranteed to carry the same amount of *FCA-mTurquoise2* transgene, see Genotyping in "Methods") (Fig. 3b–d). These data support that the m[6]A modification affects the equilibrium influencing the dynamic lifetime of the FCA nuclear condensates, promoting component association, albeit probably over short timescales in vivo.

We speculate the change in the dynamics of FCA nuclear condensates affects *COOLAIR* 3′-end processing, which underlies the de-repression of *FLC* when m[6]A is diminished. Indeed, we found a reduced use of the proximal *COOLAIR* polyadenylation site in *mta C2*, like in *fca-9* and *fy-2* (Fig. 3e). Consistently, the *COOLAIR* R-loop was increased in *mta* (Fig. 3f). Collectively, our data show that stabilization of the R-loop by NDX1 enables FCA-mediated RNA 3′-end processing to result in *COOLAIR* proximal polyadenylation and thus resolve the co-transcriptionally formed R-loop. FCA and RNA 3′-end processing factors condense into nuclear speckles and m[6]A modification enhances this condensation. Such a dynamic interplay helps explain the relationship between m[6]A and R-loop stability, which has been controversial[43,44]. We did not find global R-loop changes in *mta* (Supplementary Fig. 8), which further argues against a simple linear regulatory relationship between m[6]A and R-loop regulation, with the context being important.

**FCA and FY promote R-loop resolution at a subset of targets in the genome**. FCA had been originally identified as a regulator of flowering but was subsequently shown to have widespread roles in the *Arabidopsis* genome[45,46]. Thus, we speculated that FCA-promoted R-loop resolution might not be specific to *FLC*. Indeed, we observed R-loop accumulation globally in *fca-9* using a dot blot analysis (Supplementary Fig. 9). We had previously shown that FCA bound and promoted proximal polyadenylation of many *Arabidopsis* transcripts. The read-through transcription found in *fca* mutants generated chimeric, spliced transcripts (referred to as unannotated segments -UA loci)[46] (Supplementary Fig. 10). In order to directly test for a general association of FCA-mediated 3′-end processing with R-loop regulation, we analyzed R-loops at several UA loci. We found R-loops formed near proximal polyadenylation sites over *UA2*, *UA10B*, and *XRCC4*, which increased in *fca-9* (Fig. 4a). We also tested these R-loops in *fy-2*. The R-loops strongly increased at *UA10B* and *XRCC4* (Fig. 4b), and slightly increased at *UA2* (Fig. 4b). These data suggest that FCA-bound transcripts are frequently associated with R-loop formation and FCA/FY promotion of efficient 3′-end processing to trigger R-loop resolution occurs at least at a subset of FCA targets in the *Arabidopsis* genome.

## Discussion

In summary, our analysis identifies the importance of RNA 3′-end processing in the resolution of a *COOLAIR*-generated R-loop to trigger chromatin silencing at the *Arabidopsis* floral repressor locus, *FLC*. A simple recruitment model of the RNA-binding protein FCA to the *COOLAIR* nascent transcript is unlikely given our data. Instead, the mechanism involves dynamic nuclear condensates promoted by multivalent interactions with positive feedbacks involving m[6]A methylation. These condensates produce locally high concentrations of RNA-processing factors that efficiently terminate transcription, thus resolving the R-loop and producing the proximally polyadenylated *COOLAIR* transcript. The local chromatin environment eg. negative supercoiling, stalled RNA Pol II may be the trigger generating the R-loop structure, with the initial biochemical unwinding involving DNA/RNA helicases (Supplementary Table 1 and Fig. 4d). At *FLC*, the 3′-end processing factors in this noncanonical polyadenylation mechanism physically link to the recruitment of chromatin modifiers that silence the locus[19]. However, FCA has been shown to have widespread roles in RNA-mediated chromatin silencing in the *Arabidopsis* genome, with parallels to co-transcriptional silencing mechanisms in *S. pombe*[47,48]. This noncanonical polyadenylation mechanism could therefore be viewed as a clearing tool for chromatin "tangles" (e.g., R-loops) (Fig. 4d), frequently generated but extremely damaging to the genome, especially during DNA replication[12,49,50]. This more widespread genomic function is supported by the hypersensitivity of *fca* and *fy* mutants to the DNA damaging drug Bleomycin, as evidenced by the significant reduction in cell division measured through true leaf production in mutant versus wild-type seedlings after drug treatment (Fig. 4c and Supplementary Fig. 11a, b). Our work provides mechanistic insights on the dynamic interplay between co-transcriptionally formed R-loops, RNA-processing factors, and m[6]A RNA modification, and how that interplay connects to patterns of chromatin modification.

## Methods

**Plant materials and growth conditions**. Mutant alleles *ndx1-4*[11], *fca-9*[51], *fld-4*[8], and *fy-2*[51] were described previously. C2 line was described previously[8]. *mta* (*pABI3::MTA* in *mta*) was provided by Dr Kamil Ruzicka (Institute of Experimental Botany of the Czech Academy of Sciences) along with Dr Rupert G. Fray (University of Nottingham) and was described previously[33]. *fip37* (*pLEC1::FIP37* in *fip37-4*) was provided by Dr. Hao Yu (National University of Singapore) and was described previously[34].

To generate the *pFCA:: FCA-mTurquoise2* transgenic line, *FCA* genomic DNA was amplified and inserted into the pCambia1300 vector. The *FCA* ApaI fragment (two ApaI sites on *FCA* genomic DNA) was swapped by *FCA* ApaI fragment fused with *mTurquoise2* sequence, which was inserted before the stop codon via the HindIII site. The construct was transformed into the *fca-9* mutant.

Seedlings were surface sterilized and sown on standard half-strength Murashige and Skoog (½ MS) medium plate without glucose and stratified at 4 °C for 3 days before transferred to long-day conditions (16-h light at 20 °C, 8-h darkness at 16 °C).

**DRIP**. In all, 2-g 10-day-old seedlings were harvested and grounded into a fine powder. The powder was suspended in 30 mL of Honda buffer (20 mM HEPES, 0.44 M sucrose, 1.25% Ficoll, 2.5% dextran T40, 10 mM MgCl₂, 0.5% Triton x-100, 5 mM DTT, 1x protease inhibitor cocktail (Roche)), filtered through two layers of Miracloth, and centrifuged at 3500×g for 15 min. Nuclear pellets were resuspended in 1 mL Honda buffer and centrifuged at 8000×g for 1 min. Pellets were then resuspended in the lysis buffer (50 mM Tris-HCl pH 8.0, 10 mM EDTA, 1% SDS) supplied with 0.1 mg/mL proteinase K (AM2546, Invitrogen) and digested at 55 °C overnight with gentle rotation. The mixture was phenol/chloroform extracted, followed by DNA precipitation with NaOAc and isopropanol. The DNA pellet was dissolved gently in water and treated with Proteinase K for another 2 h, followed by phenol/chloroform extraction and DNA precipitation. The DNA pellet was dissolved in water and quantified with Qubit DNA quantification kit (Invitrogen). In total, 1 µg of DNA was dissolved in 50 µl lysis buffer (50 mM Tris-HCl pH 8.0, 10 mM EDTA, 0.5% SDS), sonicated with Diagenode Bioruptor® for 15 times, 30 s on/30 s off at high setting. DNA was then diluted ten times with dilution buffer (16.7 mM Tris pH 7.5, 167 mM NaCl, 2.2 mM EDTA, 0.1% Triton X-100) and 1% was stored at −20 °C as input. In all, 5 µg of S9.6 antibody (1:100 dilution, ENH001, Kerafast) was added, then incubated overnight at 4 °C. The next day,

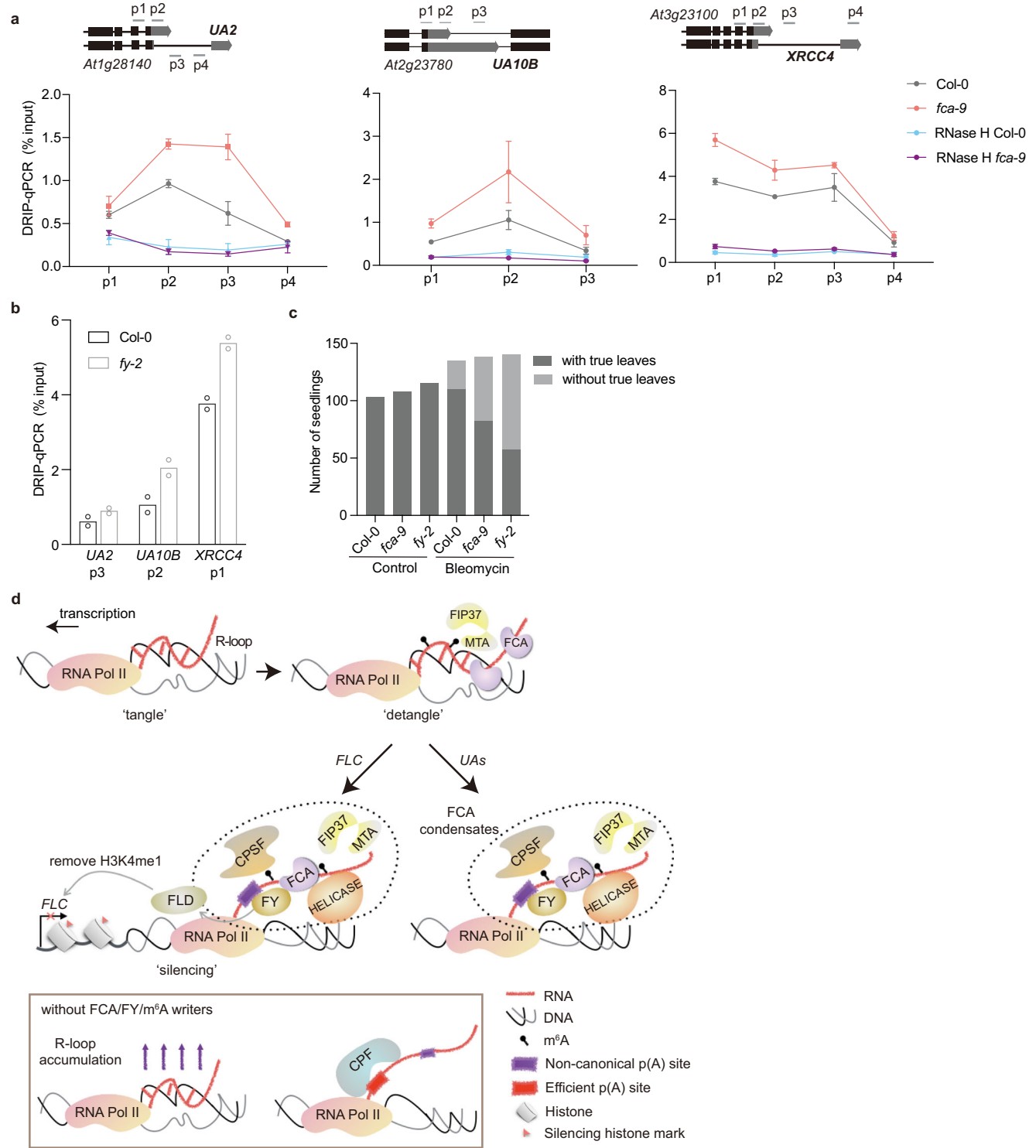

**Fig. 4 FCA and FY promote R-loop resolution at a subset of targets, so protecting the genome from DNA damage during replication. a** DRIP–qPCR analyzing R-loops at *UA2*, *UA10B*, and *XRCC4* (*UA* R-loops) in Col-0 and *fca-9*, with and without RNase H treatment. p1 to p4 are amplicons in the qPCR, which are indicated in the schematic of each locus. Data are mean ± s.d. from three biological replicates. **b** DRIP–qPCR analyzing UA R-loops in Col-0 and *fy-2*. Data are mean of two biological replicates. **c** seedlings were treated with and without bleomycin. After treatment and recovery (see "Methods"), plants with or without true leaves were scored. Data are presented as the number of seedlings in two categories. **d** "Detangle model". FCA associates with co-transcriptionally formed R-loops, potentially with Pol II stalling. FCA facilitates the deposition of m6A onto the nascent transcript by the m6A writer complex. This enhances the multivalent interactions promoting the formation of dynamic nuclear condensates. Consequently, this promotes polyadenylation at noncanonical polyadenylation (p(A)) sites, concomitantly resolving the R-loop. At *FLC*, this process is linked to chromatin silencing via interaction of the 3'-end processing factors with a histone K4 demethylase FLD. Without FCA or FY or the m6A writer complex, R-loops accumulate, and transcripts polyadenylate at efficient p(A) sites. Source data for Fig. 4a–c are provided as a Source Data file.

50 μl Protein G Agarose (Invitrogen) was added and incubated for another 2 h. The immunoprecipitants were washed five times with dilution buffer and twice with TE buffer, then were eluted in 200 μl elution buffer (10 mM Tris pH 7.5, 2 mM EDTA, 0.2% SDS, 100 ng/μl tRNA) at 55 °C for 1 h, together with input samples. The nucleic acids were precipitated with NaOAc, isopropanol, and glycogen, dissolved in water, and subjected to qPCR analysis via LightCycler480 II (Roche). The data were normalized to 1% of input. Primers were listed in Supplementary Table 2. For RNase H-treated sample, DNA was treated overnight with RNase H at 37 °C after RNase A treatment for 1 h at 37 °C.

**S9.6 dot blot**. Genomic DNA was extracted and sonicated as described in DRIP (see above). RNase H treatment was also described in DRIP. After quantification by Qubit DNA quantification kit, 2 μl DNA was blotted on Hybond™-N + membrane (Amersham). Before totally dry, the membrane was cross-linked with UV (1200 mJ/cm$^2$). The membrane was blocked in 5% (w/v) milk in TBST buffer for 1 h at room temperature, then 0.5 μg/ml S9.6 antibody (1:2000 dilution, ENH001, Kerafast) was added and incubated overnight at 4 °C overnight. After several washes in TBST buffer, the membrane was incubated in mouse IgG horseradish peroxidase linked whole antibody (1:20,000 dilution, NA931, GE Healthcare) before using chemiluminescence (Thermo Scientific) for detection. Loading was either stained in 0.1% (w/v) Methylene Blue or by SYBR™ Safe DNA Gel Stain (Thermo Scientific).

**In vivo RIP**. In all, 2-g 10-day-old seedlings were harvested and cross-linked with 1% formaldehyde. After being ground into a fine powder, the material was suspended in 30 mL of Honda buffer in the presence of 50 ng/μl RNase inhibitor (SUPERase•In™, Invitrogen) and 1x cOmplete Protease inhibitor (Roche), filtered through two layers of Miracloth, and centrifuged at $3500 \times g$ for 5 min. Nuclear pellets were resuspended in 2.5 volumes of Nuclei Lysis Buffer (50 mM Tris-HCl pH 8.0, 10 mM EDTA, 1% SDS, 1× protease inhibitor cocktail, 50 ng/μl RNase inhibitor) and sonicated with Diagenode Bioruptor® for five times, 30 s on/30 s off at high setting, followed by another ten times at low setting. Immunoprecipitation was performed by incubating 30 μl Dynabeads™ protein A (Invitrogen), FCA antibody (1:500 dilution, homemade[52]), and 1.2 mL of diluted chromatin (containing 100 μL of sonicated chromatin) at 4 °C for 1.5 h. After IP, the beads were washed four times in the washing buffer (167 mM NaCl, 16.7 mM Tris pH 7.5, 1.2 mM EDTA, 0.8% Triton X-100, 1× protease inhibitor, 50 ng/μl tRNA, 20 U/mL RNase inhibitor). Reverse cross-linking and elution were done by adding 200 μl elution buffer (2 mM EDTA, 0.2% SDS, 100 ng/μl tRNA, 0.4 U/μl RNaseOUT (Invitrogen)) to the washed beads and incubating at 55 °C overnight. The RNA from the supernatant was precipitated with isopropanol and glycogen, dissolved, DNase treated, and then used as a template for reverse transcription (RT) with gene-specific primers. Data were presented as IP/1% of input (RNA). Minus RT controls were set up to ensure the values reflect the level of RNA and not DNA contamination. UBC was used as a negative control for the experiment. Primers are listed in Supplementary Table 2.

**ChIP**. Histone ChIP was performed as previously described[53]. Protein A magnetic beads (Invitrogen, 10002D), anti-H3 (3 μg, ab1791, Abcam), and anti-H3K4me1 (6 μg, ab8895, Abcam) were used. After immunoprecipitation, recovered DNA was quantified by qPCR with primers listed in Supplementary Table 2. Data were normalized to input, and values were shown as the ratio of H3K4me1 to H3.

**Expression analysis**. Ten-day-old seedlings were harvested, and RNA was extracted. For fip37, seedlings were harvested 14 days after germination, seedlings without true leaves were taken as homozygous mutants. After treated with TURBO DNase (Ambion) to remove DNA contamination, RNA was reverse-transcribed by SuperScript IV Reverse Transcriptase (Invitrogen) using gene-specific primers. qPCR analysis was performed and data were normalized to UBC. Primers are described in Supplementary Table 2.

**Western blot analysis**. Total protein extracts were separated on NuPAGE 4–12% Bis-Tris Gels (Invitrogen) and transferred to 0.45-μm PVDF membrane (GE Healthcare). The membrane was blocked in 5% (w/v) milk in phosphate-buffered saline (with 0.1% Tween-20) (PBST) for 1 h at room temperature. Anti-FCA (1:8000 dilution, homemade) or anti-H3 (1:1500 dilution, ab1791, Abcam) antibody was added, and the incubation was carried out at 4 °C overnight. After several washes in PBST buffer, the membrane was incubated in rabbit IgG horseradish peroxidase linked whole antibody (1:10,000 dilution, NA934, GE Healthcare) before using chemiluminescence for detection. Protein loading was visualized after Coomassie Blue staining.

**Cloning, expression, and purification of recombinant proteins**. The sequence corresponding to the N-terminal of FCA (ATG to 948 bp) containing both RRM domains or FCA full length was amplified from cDNA and inserted into the pGEX-6P-1 vector (GE Healthcare). Freshly transformed cells (E. coli BL21DE3) were grown in terrific broth medium at 37 °C for 6 h, followed by induction of protein expression for 3 h at 30 °C with 1 mM IPTG. The GST-tagged protein was purified

from the cells by following a protocol provided with Glutathione Sepharose® 4 Fast Flow (GE Healthcare).

**In vitro RIP**. GST-tagged FCA protein (full length or N-terminal, including both RRM domains) was expressed and purified as described before. Human FTO protein for m$^6$A demethylation reaction was expressed and purified as previously described. For RNA extraction, nuclei were prepared as in RIP assay and digested with protease K at 55 °C for 2 h. The mixture was extracted with phenol/chloroform (pH < 5.0) followed by DNase I digestion at 37 °C for 2 h and additional phenol/chloroform extraction. For m$^6$A erased sample, 3 μg of resulting RNA were treated with 1 nmol purified FTO protein in reaction buffer (50 mM HEPES pH 7.0, 300 μM Fe$^{2+}$, 500 μM α-ketoglutaric acid, 2 mM L-ascorbic acid, 0.1 U/μl SUPERase•In, 0.5 U/μl RiboLock (Thermo Scientific™)) at 37 °C for 1 h. For the control sample, 3 μg RNA and 1 nmol purified FTO were mixed in buffer with 50 mM HEPES pH 7.0, 5 mM EDTA, 0.1 U/μl SUPERase•In, 0.5 U/μl RiboLock (without any cofactor so that the demethylation reaction never happens) incubating at 37 °C for 1 h. The m$^6$A/A concentration ratio was detected with LC-MS/MS. For each in vitro RIP sample, 150 pM protein (GST-tagged FCA-RRM or only GST) and 1.5 μg RNA (m$^6$A erased or control) were mixed in binding buffer with 20 mM Tris-HCl pH 7.5, 150 mM NaCl, 0.1% NP-40, 0.5 U/μl RiboLock at 4 °C for 2 h. Glutathione Sepharose® 4 Fast Flow (GE Healthcare) was washed and blocked with 1% BSA, 10 μg/ml yeast tRNA, and 10 μg/ml salmon sperm DNA for 2 h. For each sample, 100 μl of pre-blocked sepharose was added to the mixture incubating at 4 °C for another 2 h. The sepharose was washed with a binding buffer five times. Immunoprecipitated RNA in each sample was extracted with Trizol and reverse-transcribed using Superscript III reverse transcriptase (Invitrogen) with random hexamer. The pulldown assay using GST-tagged full-length FCA was performed under the same condition, except for using Dynabeads Protein A coated with FCA antibody instead of Glutathione Sepharose.

**Quantification of RNA modification (LC/MS-MS)**. Isolated RNA samples were digested with 0.5 U nuclease P1 in 50 μl 10 mM ammonium acetate (pH 5.3) at 42 °C for 3 h, followed by the addition of 5 μl of 1 M fresh NH$_4$HCO$_3$ and 0.5 U shrimp alkaline phosphatase (NEB). The mixture was incubated at 37 °C overnight. The resulting nucleosides were separated by UPLC and detected by Triple Quad™ 5500 (AB SCIEX) mass spectrometer. Nucleosides were quantified using the nucleoside-to-base ion mass transitions of m/z 268.0 to 136.0 (A), m/z 282.0 to 150.1 (m$^6$A). Concentrations of nucleosides in samples were calculated by fitting the signal intensities to the standard curves, and the m$^6$A/A ratios were calculated accordingly.

**FCA-binding region RNA-modification detection**. RNA immunoprecipitation was performed as previously described in RIP assay (see above), except using salmon sperm DNA instead of yeast tRNA. In total, 1% sonicated chromatin was saved as an input sample. After IP procedure, half of the washed beads were eluted and the resulting RNA was regarded as FCA–RIP sample. The other half of the beads was treated with RNase T1 (1 U/μl, Thermo Scientific) in 20 μl buffer with 20 mM Tris-HCl pH 7.5, 2.5 mM EDTA, 1× protease inhibitor cocktail at 37 °C for 10 min. The beads were then washed and eluted as described in the RIP assay to obtain the FCA-binding region sample, while the supernatant was regarded as RNase T1 digested flow-through sample. The m$^6$A/A concentration ratio was detected with LC-MS/MS for all samples as described above.

**m$^6$A immunoprecipitation**. Nuclear pellets were isolated and sonicated as described in the RIP assay (without cross-linking). RNA in the supernatant was extracted with TRIzol reagent and treated with TURBO DNase twice to remove any DNA contamination. In total, 5% extracted RNA was saved as an input sample. Each 10 μg RNA was incubated with 5 μg m$^6$A antibody (202003, Synaptic Systems) in 400 μl IP buffer (20 mM Tris-HCl pH 7.5, 150 mM NaCl, 5 mM EDTA, 0.1% NP-40, 1 × Ribolock RNase inhibitor) at 4 °C for 2 h. In all, 35 μl Dynabeads Protein A was washed three times with washing buffer (IP buffer without RNase inhibitor) and incubated with the mixture for an additional 2 h. The beads were washed five times in a new DNA LoBind tube with washing buffer. The immunoprecipitated RNA was eluted with 100 μl elution buffer (IP buffer with 10 mM N$^6$-methyladenosine) twice at 4 °C for 1 h, then precipitated with isopropanol and glycogen. The immunoprecipitated RNA and input RNA were reverse-transcribed and quantified by qPCR as described in the RIP assay. Primers are listed in Supplementary Table 2.

**Microscopy and image analysis**. Seedlings were grown on ½ MS plate with 1% (w/v) sucrose and 0.5% (w/v) phytagel (Sigma-Aldrich) for 7 days. Fresh roots were mounted in water and FCA-mTurquoise2 signal was captured by Zeiss LSM780 confocal microscope using a 40 × /1.2 water objective through the GaAsP spectral detector. mTurquoise2 was excited at 458 nm and detected at 463–579 nm. z-stack projection was applied for every image to capture 3D data of a nucleus. "StackReg" Plugin was applied for correcting drifted signal[54] when using z-stack maximum projection. The number of condensates in each nucleus was obtained manually (each counted cell was labeled and the information was available from Figshare (https://doi.org/10.6084/m9.figshare.13645730.v1). Nuclei were extracted from the

whole root for quantification (as high auto-fluorescence outside the nuclei would influence the quantification) by ImageJ software, which was used for analyzing the sizes of the condensates. "Analyze Particles" tool was applied to obtain "area" data for each condensate after manual thresholding. The same settings were used on all images). Raw data and processed images are available from Figshare (https://doi.org/10.6084/m9.figshare.13645730.v1).

**Genotyping**. C2[19], mta[33], and fip37[34] plants were genotyped, as described previously. To genotype the fca-9 mutation, genomic DNA was amplified with primers fca-9-dCAPS_F + R, respectively. PCR products were digested with StyI, followed by 3% agarose gel electrophoresis. ndx1-4 was genotyped with the transfer-DNA primer p745 and ndx1-4_GT_R to detect the presence of the insertion. PCR using ndx1-4_GT_F + R was performed to check whether the transfer-DNA insertion was homozygous. Primers are listed in Supplementary Table 2.

To obtain pFCA::FCA-mTurquoise2 in mta and appropriate control, the seeds of pFCA::FCA-mTurquoise2 homozygous plants in mta/ + background were first obtained. After germination, thirty 7-day-old seedlings were picked up, and only the root tips were mounted in water for imaging and later image analysis. After imaging, seedlings were grown for another 7 days for DNA extraction and genotyping. Among these individuals, only seven were in mta−/− background. Another seven individuals with the same genetic background (MTA + / + ) were analyzed.

**Bleomycin treatment**. Four-day-old seedlings of different genotypes were transferred to small Petri dishes containing 10 ml of the liquid growth medium, either without (control) or with 1 μg/ml of bleomycin. After 5 days of incubation (with seedlings floating in liquid, but no shaking) in the illuminated growth chamber (9-day-old seedlings), remove the medium, wash extensively by flooding the plate five times with 20 ml of liquid media. Transfer seedlings to solid medium plates. Allow seedlings to recover for 24 h before analysis. Plants were scored for true leaf emergence (shown in Supplementary Fig. 11a, b).

**Statistical analysis**. Statistical analyses were performed using software GraphPad Prism version 8.4.3. P value, sample number, and adjusted P value (after multiple comparison correction) are included in Figures or Figure legends.

**Reporting summary**. Further information on research design is available in the Nature Research Reporting Summary linked to this article.

## Data availability

The data that support this study are available from the corresponding authors upon reasonable request. Raw data and processed images are available from Figshare (https://doi.org/10.6084/m9.figshare.13645730.v1) under folders Fig. 3c and d. Source data are provided with this paper.

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

## Acknowledgements

We gratefully acknowledge Dr. R.G. Fray, Dr. K. Ruzicka, and Dr. H. Yu for providing *mta* and *fip37* mutant seeds, respectively. We thank Dr. P. Zhu and S. Reeck for the great help on image analysis. We thank Drs S. Svikovic (MRC Laboratory of Molecular Biology), S. Costa, and P. Zhu for critical reading of the paper, and C. Dean and M. Howard group members for discussions. This work was supported by the UK Biotechnology and Biological Sciences Research Council Institute Strategic Programme GEN (BB/P013511/1), the National Basic Research Program of China (2019YFA0802201 and 2017YFA0505201 for G.J.), EPSRC Physics of Life grant (EP/T00214X/1), European Research Council Advanced Investigator grant (EPISWITCH-833254), National Natural Science Foundation of China (nos. 21822702, 21820102008, and 92053109 for G.J.) and a Royal Society Professorship to C.D. Z.W. was supported by the National Natural Science Foundation of China (31970277), Key Laboratory of Molecular Design for Plant Cell Factory of Guangdong Higher Education Institutes (2019KSYS006), Guangdong Innovative and Entrepreneurial Research Team Program (2016ZT06S172), and the Shenzhen Sci-Tech Fund (No. KYTDPT20181011104005). H. D. was supported by China National Postdoctoral Program for Innovative Talent (no. BX20180007). This research was funded in part by the Wellcome Trust (210654/Z/18/Z). For the purpose of open access, the author has applied a CC BY license to any author-accepted paper version arising from this submission.

## Author contributions

C.X., Z.W., G.J., and C.D. conceived the study. C.X., Z.W., and H.D. performed all the molecular and biochemical experiments. C.X. did the genetic analysis, microscopy, and image analysis. X.F. generated *FCA-mTurquoise2* transgenic line and undertook the bleomycin experiment. C.X. and C.D. wrote the paper with all authors contributing different sections and editing the complete paper.

## Competing interests

The authors declare no competing interests.
