## [Peer Review File · Nature Communications]

REVIEWERS' COMMENTS

Reviewer #1 (Remarks to the Author):

This revised paper from the Dean lab on the role of m6A and FCA in promoting FLC chromatin silencing provides a significant new advance in understanding FLC regulated silencing. Certainly, this paper is more clearly presented and consequently, more appropriate for publication.

I do have some remaining thoughts on the paper which may help in its presentation.

1) In its present form the ms is still written in a [REDACTED] ms format. However, for Nature Comms the figures could be somewhat expanded to improve their accessibility.

2) As presented it is clear that m6A COOLAIR modification by MTA/FIP37 is required for FCA mediated FLC promoter region nucleosomal H3K4me demethylation which then allows H3K27me3 associated gene silencing. However, the exact mechanism of action of the m6A modification is as yet unresolved. What occurs to me is that possibly helicases as indicated in the model shown in Fig. 4d play a key role. Since cleavage and polyadenylation of COOLAIR by FCA and associated CPA factors is likely blocked by the R-loop structure (stabilised by NDX) possibly it is the unzipping of these R-loops that is critical for COOLAIR proximal 3' processing. What if the m6A modified RNA directly recruits such helicases? Clearly it would be exciting to test such an idea. However, this would arguably be beyond the scope of the current paper.

3) P5 13 lines down, should be Fig 1c and then 18 lines down, should be Fig. 2c.

Overall, I feel that this paper is now appropriate for publication by Nature Comms following text/figure adjustment to this journal's format. In fact, if the helicase connection as suggested above proves correct then this study would be significantly further enhanced.

Reviewer #2 (Remarks to the Author):

The authors have modified the text substantially and refocused the manuscript to specific aspects of R-loop formation. I think the revised text has addressed some of the requested experiments, and the reviewers argue that some of the requested experiments are out of the scope. With the more focused story, I think the revised manuscript is more compelling as is, and is likely to be a good candidate for

publication in Nature Communications.

Minor comments -

1. The text says "FCA-mTurquoise nuclear body number was significantly reduced" - however, the effect was fairly small, even though it was statistically significant. I think the text should state the exact % reduction in the number, size, frequency, or other metric for quantification of these nuclear bodies. This way readers won't be led to believe that these nuclear bodies largely disappear, which is what I interpreted.

2. The text and ED8 comment on the low-complexity domain of mta. It should be mentioned in the text, and perhaps by comparison in ED8, if this low complexity domain is also found in mammalian, drosophila, zebrafish mta homologs. People will wonder if these mta phase-separations could occur with other mta enzymes, or if this is likely to be unique to plants.

Reviewer #3 (Remarks to the Author):

In the revised study by Xu et al authors added new data on how NDX stabilized R-loop structure enhances FCA recruitment to COOLAIR to trigger chromatin silencing, as well as showed that FCA and 3' end processing factor mutants are hypersensitive to the DNA damage-inducing drug. Authors also totally changed the angle and the main message of the paper to focus the paper and its main message on the R-loop resolution promoting the co-transcriptional chromatin silencing.

The addition of the new data is always good. As for the shift of the papers' focus, ironically, I much preferred the previous version of the manuscript. The original manuscript was a lot more straightforward in comparison to the revised version, which becomes very convoluted towards the end. Authors at least need to re-write the paper for clarity because the mechanism is very difficult to understand. Also, it is the fact that R-loops become resolved or the process of the resolution that is important?

The Abstract is written much better but since authors examined only a few loci they need to indicate that the mechanism might operate at selected loci rather than being generic. In the abstract the authors also still need to tone down more the claim that their findings detail a mechanism because as authors say themselves in the closing sentence of the paper that there is only dynamic interplay rather than liner direct mechanistic relationship.

Minor:

--I don't know how informative the new Extended Data Fig 5 (response #6) is since the rational for using RNase T1 is still missing. Theoretically, RNase T1 cleaves after G, so unless it also cleaves after m6A the rational of using it is not clear.

- Consider change opening of the results section with "COOLAIR is ... required for..." to make it more understandable for general readership of where everything is coming from.
- Authors do not mention anywhere in the paper that their findings apply to the autonomous flowering pathway (the way I understand it) as it is often confused with the role of COOLAIR in vernalization pathway.

REVIEWERS' COMMENTS

Reviewer #1 (Remarks to the Author):

This revised paper from the Dean lab on the role of m6A and FCA in promoting FLC chromatin silencing provides a significant new advance in understanding FLC regulated silencing. Certainly, this paper is more clearly presented and consequently, more appropriate for publication. I do have some remaining thoughts on the paper which may help in its presentation.

1) In its present form the ms is still written in a [REDACTED] ms format. However, for Nature Comms the figures could be somewhat expanded to improve their accessibility.

Response: We have edited the manuscript to fulfill the Nature Comms format, including adding Introduction and Discussion sections to increase the accessibility of the topic.

2) As presented it is clear that m6A COOLAIR modification by MTA/FIP37 is required for FCA mediated FLC promoter region nucleosomal H3K4me demethylation which then allows H3K27me3 associated gene silencing. However, the exact mechanism of action of the m6A modification is as yet unresolved. What occurs to me is that possibly helicases as indicated in the model shown in Fig. 4d play a key role. Since cleavage and polyadenylation of COOLAIR by FCA and associated CPA factors is likely blocked by the R-loop structure (stabilised by NDX) possibly it is the unzipping of these R-loops that is critical for COOLAIR proximal 3' processing. What if the m6A modified RNA directly recruits such helicases? Clearly it would be exciting to test such an idea. However, this would arguably be beyond the scope of the current paper.

Response: This is a very interesting suggestion. There are only limited studies on such helicases in Arabidopsis to date. We have started to investigate candidate helicases, but so far our studies indicate there may be extensive redundancy in function. Thus, this is beyond the scope of this paper.

3) P5 13 lines down, should be Fig 1c and then 18 lines down, should be Fig. 2c.

Response: Thank you. We have corrected the mistakes.

Overall, I feel that this paper is now appropriate for publication by Nature Comms following text/figure adjustment to this journal's format. In fact, if the helicase connection as suggested above proves correct then this study would be significantly further enhanced.

Reviewer #2 (Remarks to the Author):

The authors have modified the text substantially and refocused the manuscript to specific aspects of R-loop formation. I think the revised text has addressed some of the requested experiments, and the reviewers argue that some of the requested experiments are out of the scope. With the more focused story, I think the revised manuscript is more compelling as is, and is likely to be a good candidate for publication in Nature Communications.

Minor comments -

1. The text says "FCA-mTurquoise nuclear body number was significantly reduced" - however, the effect was fairly small, even though it was statistically significant. I think the text should state the

exact % reduction in the number, size, frequency, or other metric for quantification of these nuclear bodies. This way readers won't be led to believe that these nuclear bodies largely disappear, which is what I interpreted.

Response: Thank you for the suggestion, we have edited the text adding the specific details.

2. The text and ED8 comment on the low-complexity domain of mta. It should be mentioned in the text, and perhaps by comparison in ED8, if this low complexity domain is also found in mammalian, drosophila, zebrafish mta homologs. People will wonder if these mta phase-separations could occur with other mta enzymes, or if this is likely to be unique to plants.

Response: This is a nice suggestion. Indeed, MTA orthologs in human, fruit fly and zebrafish also show a large intrinsically disordered domain, especially near the N terminus. However, they do not have a predicted prion-like domain, which is present in different plant species analyzed. This suggests the biophysical properties of MTA may differ in plants. We have included the bioinformatic analysis in Supplementary Fig 7 and edited the relevant text.

Reviewer #3 (Remarks to the Author):

In the revised study by Xu et al authors added new data on how NDX stabilized R-loop structure enhances FCA recruitment to COOLAIR to trigger chromatin silencing, as well as showed that FCA and 3' end processing factor mutants are hypersensitive to the DNA damage-inducing drug. Authors also totally changed the angle and the main message of the paper to focus the paper and its main message on the R-loop resolution promoting the co-transcriptional chromatin silencing.

The addition of the new data is always good. As for the shift of the papers' focus, ironically, I much preferred the previous version of the manuscript. The original manuscript was a lot more straightforward in comparison to the revised version, which becomes very convoluted towards the end. Authors at least need to re-write the paper for clarity because the mechanism is very difficult to understand. Also, it is the fact that R-loops become resolved or the process of the resolution that is important?

Response: We have expanded and edited the manuscript and we hope this version is more accessible. With respect to the question "it is the fact that R-loops become resolved or the process of the resolution that is important"- we hope it is now clear that it is the process of R-loop resolution by FCA/FY mediated 3'-end processing that leads to *FLC* chromatin silencing.

The Abstract is written much better but since authors examined only a few loci they need to indicate that the mechanism might operate at selected loci rather than being generic. In the abstract the authors also still need to tone down more the claim that their findings detail a mechanism because as authors say themselves in the closing sentence of the paper that there is only dynamic interplay rather than linear direct mechanistic relationship.

Response: Thanks for the suggestion. We have edited the abstract accordingly.

Minor:

--I don't know how informative the new Extended Data Fig 5 (response #6) is since the rationale for using RNase T1 is still missing. Theoretically, RNase T1 cleaves after G, so unless it also cleaves after m6A the rationale of using it is not clear.

Response: m⁶A site locates in motif [G/A/U][G>A]m⁶AC[U>A>C] , mostly in GAC. Since RNase T1 cleaves after G, it is useful to remove the residual m⁶A from the GAC context. RNase T1 is widely used to trim unprotected nucleotides, e.g., in PAR-CLIP (Hafner *et al*, 2010, Cell).

-- Consider change opening of the results section with "COOLAIR is ... required for..." to make it more understandable for general readership of where everything is coming from.

Response: We have added the "Introduction" section necessary for the Nat Comms format and to make the paper more accessible to a general reader.

-- Authors do not mention anywhere in the paper that their findings apply to the autonomous flowering pathway (the way I understand it) as it is often confused with the role of COOLAIR in vernalization pathway.

Response: Yes, our findings apply to the autonomous pathway. We previously had not introduced the autonomous pathway in order to keep the story relatively simple for those not familiar with flowering pathways. Now we have added the autonomous pathway into the "Introduction" section.